# Effect of Temperature and Light Intensity on the Polar Lipidome of Endophytic Brown Algae *Streblonema corymbiferum* and *Streblonema* sp. In Vitro

**DOI:** 10.3390/md20070428

**Published:** 2022-06-29

**Authors:** Oksana Chadova, Anna Skriptsova, Peter Velansky

**Affiliations:** A.V. Zhirmunsky National Scientific Center of Marine Biology, Far Eastern Branch of Russian Academy of Sciences, 690041 Vladivostok, Russia; askriptsova@mail.ru (A.S.); velansky.pv@gmail.com (P.V.)

**Keywords:** *Streblonema*, algae, polar lipidome, temperature adaptation, light adaptation

## Abstract

The effect of temperature and light intensity on the polar lipidome of endophytic brown algae *Streblonema corymbiferum* and *Streblonema* sp. in vitro was investigated. More than 460 molecular species have been identified in four glycoglycerolipids classes, five phosphoglycerolipids classes and one betaine lipid class. The lipids glucuronosyldiacylglycerol and diacylglyceryl-*N*,*N*,*N*-trimethyl-homoserine were found in the algae of the order Ectocarpales for the first time. A decrease in cultivation temperature led to an increase in the unsaturation level in all classes of polar lipids. Thus, at low temperatures, the content of 18:4/18:4 monogalactosyldiacylglycerol (MGDG), 20:5/18:4 digalactosyldiacylglycerol (DGDG), 18:3/16:0 sulfoquinovosyldiacylglycerol (SQDG), 18:3/18:3 and 18:3/18:4 phosphatidylglycerol (PG), 20:4/20:5 and 20:5/20:5 phosphatidylethanolamine (PE), 14:0/20:5, 16:0/20:5 and 20:5/20:5 phosphatidylcholine (PC), 20:5/20:4 phosphatidylhydroxyethylglycine and 18:1/18:2 DGTS increased. At high temperatures, an increase in the content of chloroplast-derived MGDG, DGDG and PG was observed. Both low and high light intensities caused an increase in 20:5/18:3 MGDG and 18:3/16:1 PG. At low light intensity, the content of DGDG with fatty acid (FA) 18:3 increased, and at high light intensity, it was with FA 20:5. The molecular species composition of extraplastid lipids also showed a dependence on light intensity. Thus, the content of PC and PE species with C20-polyunsaturated FA at both sn-positions, 18:1/18:1 DGTS and 16:0/18:1 phosphatidylinositol increased. Low light intensity induced a significant increase in the content of chloroplast-derived 18:1/16:1 phosphatidylethanolamine.

## 1. Introduction

Environmental factors, including temperature and light intensity, affect the growth, development and photosynthetic activity of algae. During evolution, algae have developed numerous compensatory mechanisms to smooth out the negative effects of abiotic factors. Lipid metabolism is one of the key tools in the system of adaptation of algae to changing environments [1].

Currently, many studies have been carried out on the effect of temperature on the lipid composition of algae. The main mechanisms of thermal adaptation have been determined, including changes in the degree of unsaturation and carbon chain length of fatty acids (FAs) [2,3,4], the ɷ3/ɷ6 ratio of polyunsaturated fatty acids (PUFAs) [5,6], the content of individual classes of polar lipids as well as the content of neutral lipids [7,8,9,10]. However, separate studies of FA composition and composition of lipid classes do not provide a complete picture of the adaptive reorganization of the lipid matrix of cell membranes. Increasingly popular is the lipidomic approach using HPLC-MS, which makes it possible to determine the molecular composition of all organism lipids. Currently, there are not many works devoted to the study of the effect of temperature on macroalgae lipids, these works mainly describe the composition of molecular species of polar lipids in different seasons [11,12]. However, under natural conditions, in addition to temperature, other environmental factors, such as light, salinity and the concentration of nitrogen and phosphorus, can influence the lipid composition. Recently, the influence of temperature on the molecular composition of glycolipids and betaine lipids of *Ulva lactuca* and *Saccharina japonica* cultivated under controlled conditions has been shown, which excludes the influence of other environmental factors [13,14]. Nevertheless, there is no analysis of the molecular species of phospholipids in these works.

The excess or lack of illumination affects, primarily, the activity of photosynthetic processes. The light phases of photosynthesis occur in the thylakoid membranes; therefore, it is expected that the change in the level of illumination is reflected in the FA composition of the lipids that make up these membranes. It is known that reactions to changes in light intensity are species specific, which may be due to different levels of light sensitivity of individual species, as well as differences in the degree of intensity and time of exposure to light [15]. However, a general tendency to increase the content of polar lipids and unsaturated FAs at low light intensity and to increase the content of neutral lipids and saturated FAs (SFA) at high light intensities has been observed in most studies [16,17]. Information about the effect of light on macroalgae lipidome is very limited. Currently, only one study has been carried out at the lipidome level on the example of green macroalgae [18].

The objects of our study are filamentous endophytic algae *Streblonema corymbiferum* and *Streblonema* sp. (Ectocarpales, Phaeophyceae), isolated from *Eualaria fistulosa* (Laminareales, Phaeophyceae). These basiphytes are mainly distributed in the North Pacific and Bering Sea [19]. Endophytes *S. corymbiferum* and *Streblonema* sp. are poorly studied and information on their biochemical composition is almost completely absent. These algae have a high growth rate and can be cultivated under controlled conditions in photobioreactors, which makes them convenient objects for research and promising sources of bioactive substances [20]. The aim of this study was to determine the effect of temperature and light intensity on the composition of molecular species of polar lipids in *S. corymbiferum* and *Streblonema* sp. and to establish the main mechanisms of thermal and light adaptations in vitro.

## 2. Results

### 2.1. The Polar Lipidome of Streblonema corymbiferum and Streblonema sp.

A total of 10 lipid classes were identified, including glycoglycerolipids (GL)—mono- and digalactosyldiacylglycerol (MGDG and DGDG), sulfoquinovosyldiacylglycerol (SQDG) and glucuronosyldiacylglycerol (GlcADG); phosphoglycerolipids (PL)—phosphatidylglycerol (PG), phosphatidylethanolamine (PE), phosphatidylcholine (PC), phosphatidylinositol (PI) and phosphatidylhydroxyethylglycine (PHEG); and betaine lipid (BL)—diacylglyceryl-*N*,*N*,*N*-trimethyl-homoserine (DGTS) (Table 1). The composition of MGDG and DGDG was dominated by molecular species with C18 PUFA in both positions (C18-PUFA/C18-PUFA) and C20-PUFA/C18-PUFA. Among glycolipids, SQDG was characterized by the highest degree of saturation. The major molecular species of SQDG included 14:0, 16:0, 16:1 and C18 FA. GlcADG mainly contained 16:0, 20:5 and C18 FA (quantitative analysis of the molecular species composition of GlcADG was not carried out, due to the extremely low content). PG predominantly contained C18 FA with varying degrees of saturation, as well as 16:0 and 16:1 FA. The major molecular species of PE contained 20:4 and 20:5 FA at one or both positions of the glycerol backbone. The composition of the molecular species of PC was similar to that of PE, due to their common biosynthetic pathway. The main molecular species of PI predominantly contained 14:0, 16:0 and 18:0 FA at the sn-1 position, and C18 FA with varying degrees of saturation at the sn-2 position. PHEG had only two major molecular species—20:4/20:5 and 20:4/20:4. Betaine lipid DGTS predominantly contained 16:0 and 18:1 FA.

It should be noted that we did not find lysoforms of any polar lipids.

### 2.2. Effect of Temperature on the Polar Lipidome of Streblonema corymbiferum and Streblonema sp.

The comparative analysis of the content of molecular species of polar lipids, carried out in the present work, showed a distinct dependence of the unsaturation level on temperature. Significant changes were observed both in the composition of the plastid membranes lipids (MGDG, DGDG, SQDG and PG) and in the composition of the structural components of the extraplastid membranes PC, PE and DGTS (Appendix A).

The content of PUFA/PUFA and SFA/PUFA MGDG molecular species was maximum at low temperature (5 °C). With increasing temperature, the content of molecules with monounsaturated acids (MUFA), PUFA/MUFA and MUFA/PUFA, increased. The SFA/MUFA level was highest at the medium temperature (15 °C). The content of MGDG molecular species with extremely unsaturated fatty acids (18:4/18:4 and 20:5/18:4 MGDG) was maximum at low temperatures (5 °C), with less unsaturated FA (18:3/18:4 and 18:3/18:3) at medium temperatures (15–20 °C), with the least unsaturated FA (18:2/18:4, 18:3/18:2, 18:1/18:3, 20:5/ 18:2 in both endophytes and 20:5/18:1 in *S. corymbiferum*) at high temperatures (25 °C) (Figure 1). Increased temperature also induced an increase in 20:4/20:4 content in both endophytes. With increasing temperature, the percentage of 20:5/18:3 in *S. corymbiferum* increased (maximum value at 25 °C), while in *Streblonema* sp. its level was maximum at a medium temperature.

Similar to MGDG, levels of PUFA/PUFA DGDG molecular species were highest at a low temperature (5 °C) and PUFA/MUFA were highest at a high temperature (25 °C). At low temperatures, the content of SFA/MUFA decreased. With decreasing temperature, the content of the most unsaturated DGDG species (20:5/18:4) increased, and with increasing temperature, the content of less unsaturated molecular species (20:4/18:3, 20:5/18:2, 20:5/18:1 in both endophytes and 18:3/18:2 in *Streblonema* sp.) increased (Figure 2). Similar to MGDG, the percentage of 20:5/18:3 DGDG increased in *S. corymbiferum* with a maximum value at 25 °C, while in *Streblonema* sp. the maximum content of this species was observed at 15 °C.

The amount of SFA/SFA and SFA/MUFA SQDG molecular species was highest at high temperatures. The content of MUFA/SFA was maximum at medium temperatures. The percentage of molecular species containing PUFA increased at low temperatures. The content of the most unsaturated SQDG molecular species (18:3/16:0 and 14:0/18:3) increased at lower temperatures, and the content of the most saturated species (14:0/16:0, 16:0/16:0, 14:0/18:1, 16:0/16:1, 16:1/16:0) increased at high temperatures (Figure 3). The percentage of 18:1/16:0 molecular species (with 18:1 *n*-9 FA, presumably) had a maximum value at medium temperatures, and the percentage of 18:1/16:0 isomer (with 18:1 *n*-7 FA, presumably), apparently distinguished by the double bond position at 18:1, was highest at low temperatures. The 18:2/16:0 content in *S. corymbiferum* was highest at low temperatures, while in *Streblonema* sp. it was highest at high temperatures.

With decreasing temperature, the content of the most unsaturated PG species (18:3/18:4 and 18:3/18:3) increased, while with increasing temperature, the level of the less unsaturated species (18:3/16:0 and 18:1/18:1) increased (Figure 4). The content of 16:0/18:1 was maximum at medium temperatures, while the total amount of 16:0/18:2 and 18:2/16:0 isomers, on the contrary, was minimal at these temperatures.

The content of PUFA/PUFA PE molecular species was maximum at low temperatures, while SFA/PUFA percentage increased at high temperatures. At low temperatures, the content of the most unsaturated PE molecular species (20:4/20:5 and 20:5/20:5) increased, while at high temperatures the content of the 20:4/20:4 species increased (Figure 5). The amount of the most saturated PE molecular species (14:0/20:4, 16:0/20:4 and 22:0/20:4 in both endophytes and 14:0/20:5 in *S. corymbiferum*) increased with increasing temperature.

The pattern of thermal adaptation changes in PC was more complex compared to that of PE. A decrease in temperature induced an increase in the percentage of PUFA/PUFA and SFA/PUFA molecular species. At high temperatures, the levels of SFA/MUFA and PUFA/MUFA molecular species increased. At lower temperatures, the content of the most unsaturated molecular species (20:5/20:5) increased (Figure 6). Low temperatures also induced an increase in the content of species containing 20:5 FA (16:0/20:5, 20:5/18:3 in both endophytes, 14:0/20:5 in *Streblonema* sp.), whereas high temperatures increased the content of PC molecular species with less unsaturated FA (14:0/18:1, 14:0/18:2, 16:0/18:1, 16:0/18:2, 20:4/18:1 in both endophytes, 14:0/20:4, 16:0/20:4, 20:4/18:2 in *Streblonema* sp.).

The PI molecular composition was almost unaffected by temperature. High temperatures induced an increase in the most saturated PI molecular species (16:0/18:0, 14:0/18:1, 16:0/16:1) only in *S. corymbiferum* (Figure 7).

At low temperatures, the percentage of the more unsaturated PHEG species (20:5/20:4) increased, and at high temperatures, the content of the less unsaturated species (20:4/20:4) increased (Figure 8).

At low temperatures, the content of the most unsaturated DGTS molecular species (18:1/18:2) increased, the content of the less unsaturated species (18:1/18:1) was highest at medium temperatures, and the content of the least unsaturated species (16:0/18:1) was highest at high temperatures (Figure 9). The percentage of unresolved isomers 16:0/20:1 + 18:1/18:0 was higher at low and high temperatures compared to medium temperatures.

### 2.3. Effect of Light Intensity on the Polar Lipidome of Streblonema corymbiferum and Streblonema sp.

Changes in the composition of polar lipid classes under the influence of light intensity were less pronounced and more complex than those under the influence of temperature (Appendix A). The content of the most unsaturated MGDG molecular species (18:4/18:4, 20:5/18:3 and 20:5/18:4) increased at both low and high light intensities, except for 20:5/18:4 in *Streblonema* sp., whose level decreased at light intensities above 20 μmol photons m^−2^ s^−1^ (Figure 10). The content of less unsaturated molecular species (18:3/18:2 and 18:2/18:4) was maximum in the range of medium light intensity (20–50 μmol photons m^−2^ s^−1^). At high light intensity, the percentage of the least unsaturated MGDG molecular species (16:0/18:3 and 16:0/18:1) increased in both endophytes. The amount of the 18:3/18:3 molecular species also increased at high light intensity.

The content of 18:3/18:2, 18:3/18:3 and 18:3/18:4 DGDG molecular species was maximum in the absence of illumination, while 20:5/18:4 was maximal at high light intensity (Figure 11). The 20:5/18:3 amount increased both at low light intensity and at high light intensity, while 20:4/18:4, on the contrary, was maximum at medium light intensity.

There were no significant changes in the SQDG composition depending on the light intensities. At high light intensities, the content of 18:2/16:0 molecular species decreased (Figure 12).

At high light intensity, the content of the most saturated PG molecular species (16:0/18:1 and 18:1/18:1) increased (Figure 13). The percentage of 18:3/16:1 was higher at low and high light intensities compared to the medium light intensities. In contrast, the percentage of 18:3/16:0 was maximal at medium light intensities.

High light intensity induced an increase in the content of highly unsaturated C20/C20 PE molecular species (20:4/20:5 in both endophytes, 20:5/20:5 in *S. corymbiferum*), while low light intensity induced an increase in the content of molecular species with C16 and C18 FA (18:1/16:1, 18:3/20:5, 20:5/18:4 in both endophytes, 14:0/20:4 in *S. corymbiferum* and 16:0/20:4 in *Streblonema* sp.) (Figure 14). The amount of 20:4/20:4 molecular species decreased at low light intensity.

The content of the most unsaturated PC molecular species (20:4/20:5 and 20:5/20:5) was maximal at high light intensity, whereas the percentage of 20:4/20:4 was maximal at medium light intensity (50 μmol photons m^−2^ s^−1^) (Figure 15). The 16:0/20:5 content in both endophytes and 14:0/20:5 in *S. corymbiferum* was minimal at medium light intensity (12–50 μmol photons m^−2^ s^−1^). The levels of the least unsaturated molecular species (14:0/18:1 and 16:0/18:1) was maximal at light intensity 5–20 μmol photons m^−2^ s^−1^. The 20:4/18:1 content in *Streblonema* sp. was maximal at low light intensity.

At low light intensity, the level of the 16:0/18:2 PI slightly increased, and at high light intensity, the content of 16:0/18:1 slightly increased (Figure 16).

There were no significant changes in the PHEG composition depending on the light intensity.

At low light intensities, the amount of the most unsaturated DGTS molecular species (18:1/18:2) slightly increased, while at high light intensities the level of the less unsaturated species (18:1/18:1) increased (Figure 17). At low light intensity, the sum of 16:0/20:1 and 18:1/18:0 isomers also increased.

## 3. Discussion

### 3.1. The Polar Lipidome of Streblonema corymbiferum and Streblonema sp.

The glycoglycerolipids MGDG, DGDG and SQDG and the phosphoglycerolipids PG, PE, PC, PI and PHEG identified in endophytes are common in brown algae. The high content of PUFA with 18 and 20 carbon atoms is also a characteristic feature of brown algae [21] (pp. 47–64). Glycoglycerolipid GlcADG and betaine lipid DGTS were found in the algae of Ectocarpales for the first time. The presence of GlcADG was previously established in higher plants [22], unicellular algae [23], sea grasses [24] and in some bacteria and fungi [25]. Previously, we identified this lipid in the brown algae *Undaria pinnatifida* (Laminareales) [26]. It is known that in higher plants GlcADG is synthesized in chloroplasts from UDP-glucuronic acid (UDP-GlcA) and diacylglycerol (DAG) by SQDG synthase (SQD2) [22]. In addition, the authors found that the FA composition of GlcADG and SQDG was similar, which also indicated a common biosynthetic pathway for these lipids. Our results indicate that the molecular species composition of GlcADG is similar to MGDG and DGDG. Thus, two pathways of GlcADG biosynthesis in brown algae can be suggested: (1) synthesis from DAG and UDP-GlcA by SQDG synthase, MGDG can be a possible DAG source; (2) synthesis directly from MGDG by modifying the polar head group. The betaine lipid DGTS is common in green algae, but it is generally rare in brown algae [21] (pp.118–128). Previously, another BL DGTA was found in Ectocarpales algae [27]. The major FA in DGTS composition of endophytes was 18:1. DGTS probably acts as the primary substrate for FA desaturation in the endoplasmic reticulum (ER) [28].

### 3.2. Effect of Temperature on the Polar Lipidome of Streblonema corymbiferum and Streblonema sp.

The ability of algae to survive changes in environmental temperature indicates the presence of mechanisms to cope with temperature stress. One of the most studied mechanisms of adaptation of organisms to changes in environmental temperature is a change in the degree of unsaturation of membrane lipids, aimed at maintaining the optimal level of membrane fluidity by regulating the activity of desaturases [29]. The obtained results indicate that the degree of contribution of different classes of lipids to the process of adaptation to low and high temperatures is different. In general, the relationship between saturation level and temperature was observed in all classes of polar lipids of endophytes.

Changes in the molecular composition of thylakoid lipids (MGDG, DGDG, SQDG, GlcADG and PG) are of the greatest interest since their function is not only to maintain the membrane structure, but they also play an important role in the photosynthetic process [30]. The FA composition of lipids is important for the stabilization of photosynthetic proteins in the membrane under thermal stress [31]. MGDG and DGDG, the main components of thylakoid membranes, showed a similar reaction in response to temperature changes. The decrease in temperature was accompanied by an increase in the content of MGDG and DGDG molecular species with extremely unsaturated fatty acids: 18:4/18:4 and 20:5/18:4. Adaptation to low temperatures in the SQDG composition was accompanied by an increase in the content of the most unsaturated molecular species with FA 18:3: 18:3/16:0 and 14:0/18:3. At low temperatures, PG unsaturated species 18:3/18:3 and 18:3/18:4 also accumulated. PUFAs are highly structurally flexible, allowing the lipid to conform to the various forms of numerous photosynthetic proteins. An increase in the content of highly unsaturated molecular species of thylakoid lipids at low temperatures is necessary to maintain the fluidity of thylakoid membranes and, consequently, the efficiency of photosynthesis. In addition, under chilling-induced photoinhibition, the degree of unsaturation of thylakoid membrane lipids affects the rate of photosystem II (PS II) repair, during which the damaged D1 protein is replaced [32,33,34]. High temperature induced an increase in the content of less saturated species of all classes of thylakoid lipids. Thus, the number of MGDG and DGDG molecular species mainly with 18:3, 18:2 and 18:1 FAs at the sn-2 position increased, while the content of species with FA 18:4 decreased. Therefore, thermal adaptation in the composition of galactolipids occurred due to a change in the unsaturation degree of C18 FA. At high temperatures, the level of PG and SQDG molecular species with MUFA and SFA increased. These FAs reduce the fluidity of the membrane bilayer, which, in turn, ensures the efficiency of photosynthetic processes under these conditions.

Phosphoglycerolipids PC, PE, PI, PHEG are structural components of extraplastid membranes. PC is the most diverse class of lipids in terms of composition, it is an intermediate in the synthesis of other phosphoglycerolipids. PC is present in the chloroplast outer envelope membrane and can be a FA donor for glycolipid synthesis [35]. It seems that 20:5 FA in the PC and PE composition plays the key role in maintaining the required level of membrane fluidity at low temperatures. With an increase in temperature, an increase in the PC and PE molecular species containing SFA and MUFA, which ensures a decrease in the molecular mobility of the lipid bilayer, should lead to the resistance of cells to high temperatures. The ratio of 20:5 and 20:4 FAs in PHEG, as well as in PC and PE, varied depending on the temperature.

A slight change in the content of DGTS molecular species indicates the secondary participation of this lipid in the processes of temperature adaptation. However, the dependence of the unsaturation degree of this lipid on temperature was also observed. Thus, at low temperatures, the 18:1/18:1 major molecular species desaturates at the sn-2 position to form 18:1/18:2. Under the influence of high temperatures, the activity of desaturases decreased.

It is known that plants are capable of regulating the FA unsaturation level by changing the degree of contribution of the prokaryotic and eukaryotic pathways in lipid synthesis [36]. In plant cells, DAG assembly occurs in both the ER (eukaryotic pathway) and chloroplasts (prokaryotic pathway), while de novo FA synthesis only occurs in chloroplasts. For DAG synthesis in the ER, free FAs are transported from chloroplasts presumably by the FAX 1 protein [37]. Due to the substrate specificity of lysophosphatidic acyltransferases, plastid-derived DAGs contain C16 FA at the sn-2 position, while ER-derived DAGs contain C18 FA. DAGs synthesized in the ER can be transported into the chloroplast envelope by the TGD1-5 protein complex and used for the synthesis of thylakoid lipids [38]. Synthesis of C20 FA in algae occurs in the ER, then they can be transported to chloroplasts [39].

MGDG and DGDG *S. corymbiferum* and *Streblonema* sp. were predominantly of ER origin, since most molecular species contained C18 and C20 FA at the sn-2 position (96.0% and 98.1% at 15 °C, respectively). A decrease in temperature had almost no effect on the ratio of eukaryotic and prokaryotic galactolipids (Figure 18). With an increase in temperature, the contribution of the prokaryotic pathway to the biosynthesis of MGDG and DGDG slightly increased. SQDG is the only lipid of *S. corymbiferum* and *Streblonema* sp. which is synthesized predominantly in chloroplasts (90.0% and 92.8% at 15 °C, respectively). The change in temperature did not influence the origin of this lipid. At low temperatures, the level of the ER-derived PG increased, which may be due to the limited FA desaturation in chloroplasts. Thus, in chloroplasts, the maximum number of double bonds that can be formed in an FA chain at sn-2 position is one, and in ER it is five. An increase in the content of ER-derived PG at low temperatures was previously described in wheat [36]. At high temperatures, the amount of chloroplast-derived PG increased, mainly through 18:3/16:0 molecular species; however, at the same time, the level of ER-derived PG with 16:0, 18:1 and 18:2 FA also increased. This also indicates the involvement of both biosynthetic pathways in the process of adaptation to heat stress.

As a result of the analysis, plastid-derived molecular species of extraplastid lipids, such as PC, PE and DGTS, were found. The presence of such lipids suggests the possibility of DAG transport for their synthesis from chloroplasts. The amount of PC and PE chloroplast-derived molecular species was minor (less than 1.5%) and did not change under the influence of temperature, while the amount of DGTS chloroplast-derived molecular species did not exceed 10% of the sum of all molecular species.

### 3.3. Effect of Light Intensity on the Polar Lipidome of Streblonema corymbiferum and Streblonema sp.

Light is a critical environmental factor affecting algae metabolism. Due to the lack of solar energy, the efficiency of photosynthesis can decrease, which causes the development of algae to slow down [40]. Light stimulates photosynthesis, but with excessive light intensity, an imbalance between the absorbed energy and the ability to use it can occur, resulting in oxidative stress, accompanied by the destruction of chloroplasts [41]. To maintain photosynthetic activity at an optimal level and protect photosystems, various adaptation mechanisms have been developed in algae. Modifications in the lipid composition of endophytes that occur with changes in light intensity are probably also adaptive responses. At low light intensity, an increase in the content of the most unsaturated MGDG species, containing predominantly 20:5 and 18:4 FA, may be necessary to increase membrane fluidity and, consequently, the rate of photosystem electron transport [42]. Previously, it was shown that the accumulation of 20:5 *n*-3 in MGDG under low light conditions in *U. pinnatifida* is accompanied by an increase in the chlorophyll content, which together leads to an increase in photosynthetic activity [43]. In addition, FA desaturation of plastid lipids induces a change in lipid–protein interactions, which, in turn, affects the self- assembly of active chlorophyll–protein complexes [44]. At low light intensity, the amount of DGDG containing 18:3 FA at the sn-1 position also increased. At high light intensity, an increase in some highly unsaturated molecular species of MGDG and DGDG may be related to both the structural and protective function of these lipids. It is known that MGDG is involved in the xanthophyll cycle, a key mechanism for protecting the photosynthetic apparatus of plants from damage caused by excessive light exposure. The presence of highly unsaturated non-bilayer MGDG is a necessary condition for carotenoid solubilization and activation of violaxanthin de-epoxidase, which catalyzes the formation of the photoprotector zeaxanthin [45]. Previously, it was shown that PSII repair is inhibited in DGDG-deficient mutants under high light conditions compared to the wild type [46]. It is likely that highly unsaturated DGDG (20:5/18:4 and 20:5/18:3) may be necessary to maintain the rate of resynthesis of the D1 protein of PSII. An increase in the unsaturation level of galactolipids may also be associated with the PUFA ability to neutralize reactive oxygen species and thus perform a protective function [47]. At low and high illuminations, the content of the 18:3/16:1 molecular species increased in the PG composition. We suggest that 16:1 FA at the sn-2 position is a 16:1Δ3-trans PG-specific FA, inducing an increase in the light-harvesting complex II (LHCII) trimerization [48]. Under low levels of illumination, this may be necessary to increase the light-harvesting ability of PSII. The increase in the number of LHCII trimers at high light intensity is probably associated with a photoprotective function.

Changes in light intensity also affected the composition of extraplastid lipids. Low light intensity induced an increase in the content of the 18:1/16:1 PE molecular species, indicating increased transport of DAG from chloroplasts to the ER for the synthesis of this lipid. This was especially noticeable in *Streblonema* sp., where the content of this species was 0.3% at the maximum level of illumination and 26.3% in the absence of light. It is also interesting that in other classes of polar lipids, the 18:1/16:1 species occur in trace amounts.

At high light intensity, an increase in the content of molecular species with C20 PUFA in PC and PE was noted, which primarily indicates an increase in FA synthesis and can also be associated with the antioxidant function of PUFA.

In DGTS, with increasing light intensity, the amount of 18:1 FA increases (approximately from 60 to 80% of all DGTS acyl groups in both endophytes), and the amount of the 18:1/18:1 DGTS increases from 40 to 60–70%. This most likely indicates an increase in de novo FA synthesis and confirms the role of DGTS as a substrate for the primary FA extraplastid desaturation. The amount of 18:1 FA and the 16:0/18:1 species also increases in another extraplastid lipid PI.

## 4. Materials and Methods

### 4.1. Algae material

*S. corymbiferum.* and *Streblonema* sp. were isolated from a culture of gametophyte clones of *Eualaria fistulosa* (Laminareales, Phaeophyceae) (Bering Island). Basiphyte (host algae) thalli fragments containing endophytes were placed in Petri dishes with sterilized (ultraviolet, 0.2 μm filtration, boiling) seawater enriched with ES medium [49] and kept at a temperature of 15 ± 1 °C, a medium light intensity of 10 μmol photons m^−2^ s^−1^ and a photoperiod of 12 h light:12 h dark for spore release. After 2 days, the basiphyte fragments were removed. The filaments of endophytes were removed from culture and grown separately in sterilized (ultraviolet, 0.2 μm filtration, boiling) natural seawater fertilized with an ES medium [49]. The cultures were illuminated with cool light fluorescence lamps (Phillips 39 W) that provided a medium light intensity of 25–30 μmol photons m^−2^ s^−1^. The photoperiod was 12 h of light and 12 h of dark.

Endophytes were identified using descriptions of filamentous ectocarpalean algae [50,51,52,53,54,55]. Identification images of endophytes and information on species identification are provided in Appendix A.

### 4.2. Temperature Treatment

Samples were grown in 200 mL flasks with sterile seawater enriched with ES medium [49] for three weeks at temperatures of 5, 10, 15, 20 and 25 °C, a medium light intensity of 200 μmol photons m^−2^ s^−1^ and a photoperiod of 12 h light:12 h dark. Three samples of each endophyte species were cultivated in each temperature regime.

### 4.3. Light Treatments

Samples were grown in 200 mL flasks with sterile seawater enriched with ES medium [49] for three weeks at light intensities of 0, 5, 12, 20, 50, 100, 150 and 200 μmol photons m^−2^ s^−1^, temperature of 14–16 °C and a photoperiod of 12 h light:12 h dark. Three samples of each endophyte species were cultivated in each light regime.

### 4.4. Lipid Extraction

Samples of endophytes were collected on filter paper, dried, weighed and transferred to glass tube. Lipids were extracted using the mixture of chloroform: methanol (1:1). The organic phase was collected in pear-shaped flasks using filter paper. The biomass was re-extracted six times. The final organic phase was dried by rotary evaporator, transferred to glass vials, dried, weighed and stored in chloroform at −20 °C.

### 4.5. HPLC-MS/MS Analysis of Molecular Species

Separation of polar lipid molecular species was performed using Shimadzu HPLC system (Kyoto, Japan), equipped with degassing units (DGU-20A3r and DGU-20A5r), four pumps (LC-30AD), an autosampler (SIL-30AC), a column oven (CTO-20AC) and a controller CBM-20A. Column Ascentis Express C18 (150 × 2.1 mm i.d., 2.7 μm) (Supelco, Bellefonte, USA) was operated at 70 °C. Mobile phases were used: A, methanol; B, 2-propanol; C, water containing 2 M formic acid and 1.8 M ammonia; D, water. A, B and D eluent channels were connected to the mixer (40 mkl volume) through cartridge (10 mm × 2 mm ID) with SCX-1001 cationite (Yanaco, Japan) and the C channel was connected directly to a mixer. Eluent was pumped at a constant flow 0.2 mL min^−1^ with gradient (A:B:C:D, % by vol.): 0 min (33.75:41.25:2.5:22.5), 5 min (28.5:46.5:2.5:22.5), 15 min (24.75:50.25:2.5:22.5:), 20 min (11.25:63.75:2.5:22.5), 22 min (0:75:2.5:22.5), 30 min (0:82.5:2.5:15), 45 min (0:90:2.5:7.5), 49 min (0:100:0:0), 55 min (0:100:0:0) and 55.01–61 min (33.75:41.25:2.5:22.5). In total, 0.2–0.5 mkl of lipid extract with 1 mg/mL concentration was injected.

Quantitative analyses of molecular species and their identification using a fragmentation pattern were performed on a Shimadzu LCMS-8060 triple-quadrupole mass spectrometer (Kyoto, Japan) with electrospray ionization (ESI) ion source. The temperature of the interface, heat block and desolvation line was 300, 400 and 250 °C, respectively. The flow rates of drying gas (N_2_), nebulizer gas (N_2_) and heating gas (zero air) were 10 L min^−1^, 3 L min^−1^ and 10 L min^−1^, respectively. The negative ion mode was applied for quantitative analysis of PI and the positive mode was applied for analyzing others lipid classes. Previously described fragmentation reactions were used for sn-position of acyl chains determination in all polar lipid classes [56,57,58,59,60], except GlcADG and PHEG. For the identification of MGDG, DGDG and PC, [M + Li]^+^ ions were used, in this case 5mM LiOH in methanol with 0.02 mL/min flow was added postcolumn. Mass spectrometer parameters (precursor and fragment ions, collision energy and type of registered fragmentation reaction) for qualitative and quantitative analyses of each class of polar lipids are given in Appendix A. MS/MS spectra and fragmentation reactions are shown in Appendix A.

### 4.6. Statistical Analysis

All statistical analyses were performed using Microsoft Excel (Microsoft, Redmond, WA, USA). All values were presented as mean ± standard deviation for triplicate. The data were assessed statistically by one-way ANOVA and Tukey HSD test for a posteriori comparisons. A probability level of *p* < 0.05 was considered significant.

## 5. Conclusions

The obtained results indicate that changes in temperature affected the composition of all classes of lipids to varying degrees. At low temperatures, the content of extremely unsaturated (possible for the particular polar lipid class) molecular species of lipids (MGDG and DGDG with 18:4 and 20:5, SQDG with 18:3, PG with 18:3 and 18:4, PC, PE and PHEG with 20:5, PI and DGTS with 18:2) increased. At high temperatures, the amount of these molecular species decreased, while the content of other (less unsaturated) species increased. Such modifications are necessary to maintain the fluidity of cell membranes at an optimal level. Temperature had the least significant effect on the composition of PI and PHEG.

With changes in illumination, the modifications of the plastid lipid composition are mainly aimed at maintaining the function of the photosynthetic apparatus. Light intensity had a significant effect on the composition of plastid membrane lipids, with the exception of SQDG. Both low and high light intensities induced an increase of 18:3/16:1 PG amount. With increasing light intensity, the amount of C20-PUFAs (especially 20:5) in MGDG and DGDG increased, which is associated with the role of these lipids in the photosynthetic apparatus of chloroplasts. The change in light intensity had little effect on the SQDG molecular composition; apparently, this lipid is not involved in light adaptation. Changes in the extraplastid lipid composition are probably the result of increased de novo FA synthesis, which leads to the accumulation of 18:1 FA in PI and DGTS and the end products of FA synthesis in brown algae—C20-PUFA in PC and PE.

The study of the contribution of pro- and eukaryotic pathways of synthesis revealed an increase in the level of plastid-derived molecular species in MGDG, DGDG and PG at high cultivation temperatures. This additional mechanism allows us to regulate the level of unsaturated lipids, which is presumably related to the peculiarities of desaturation in ER and plastids. A significant increase in plastid-derived PE 18:1/16:1 under low light conditions indicates the presence of a DAG transport pathway from chloroplasts. In general, the high level of unsaturation of most classes of lipids under optimal conditions indicates the cold and light resistance of endophytes. On the other hand, this may indicate an increased sensitivity to high temperatures [61].

It is also worth noting that despite the close relationship of the two species of endophytes, the quantitative composition of some molecular types of lipids and adaptation mechanisms differed in them.

Thus, the analysis of the polar lipidome, including the separation of isobaric molecular species and the determination of the sn-position of acyl chains, made it possible to obtain a complete picture of the adaptive changes that occur in brown endophytic algae during cultivation under various temperature and illumination conditions.

## Figures and Tables

**Figure 1 marinedrugs-20-00428-f001:**
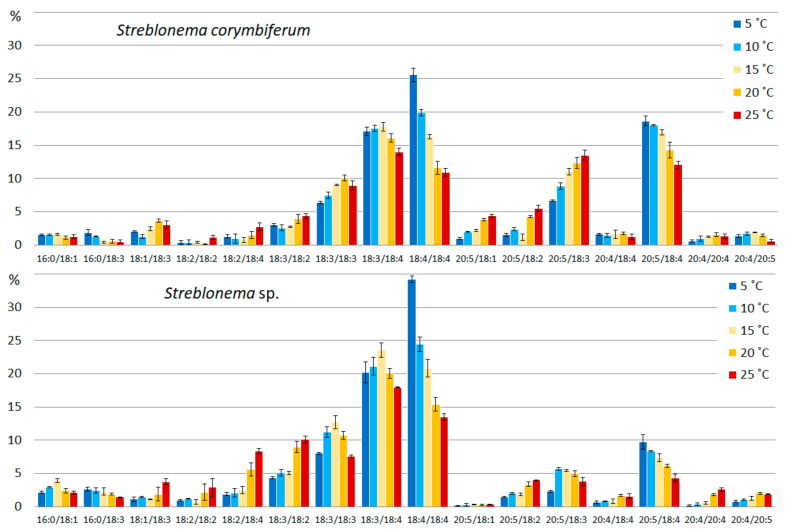
Monogalactosyldiacylglycerol molecular species composition (% of total MGDG) of *Streblonema corymbiferum* and *Streblonema* sp. at various temperatures.

**Figure 2 marinedrugs-20-00428-f002:**
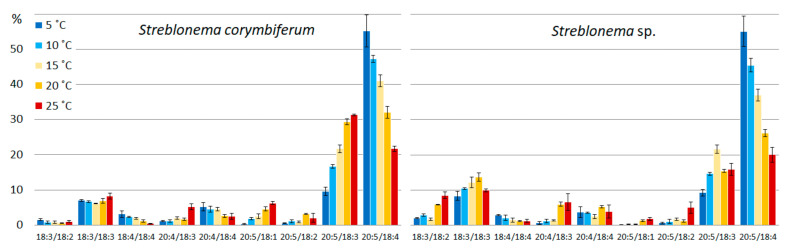
Digalactosyldiacylglycerol molecular species composition (% of total DGDG) of *Streblonema corymbiferum* and *Streblonema* sp. at various temperatures.

**Figure 3 marinedrugs-20-00428-f003:**
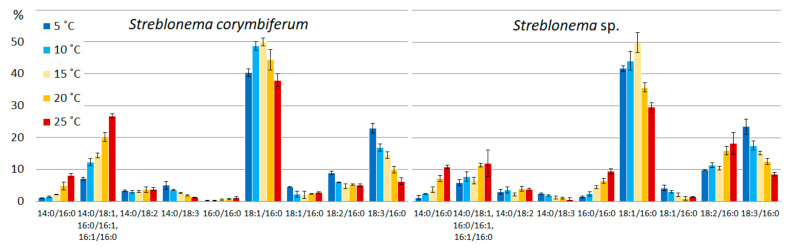
Sulfoquinovosyldiacylglycerol molecular species composition (% of total SQDG) of *Streblonema corymbiferum* and *Streblonema* sp. at various temperatures.

**Figure 4 marinedrugs-20-00428-f004:**
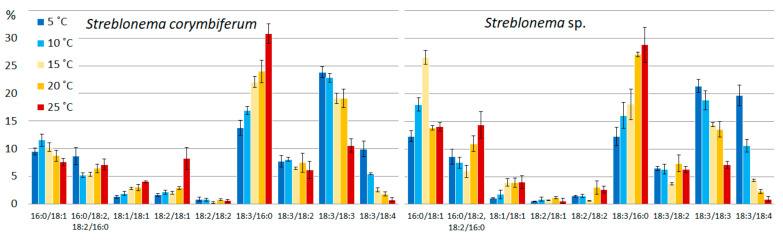
Phosphatidylglycerol molecular species composition (% of total PG) of *Streblonema corymbiferum* and *Streblonema* sp. at various temperatures.

**Figure 5 marinedrugs-20-00428-f005:**
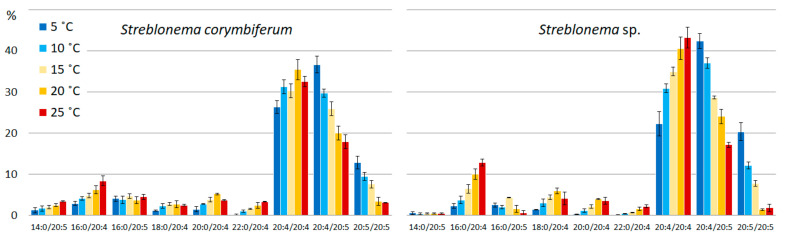
Phosphatidylethanolamine molecular species composition (% of total PE) of *Streblonema corymbiferum* and *Streblonema* sp. at various temperatures.

**Figure 6 marinedrugs-20-00428-f006:**
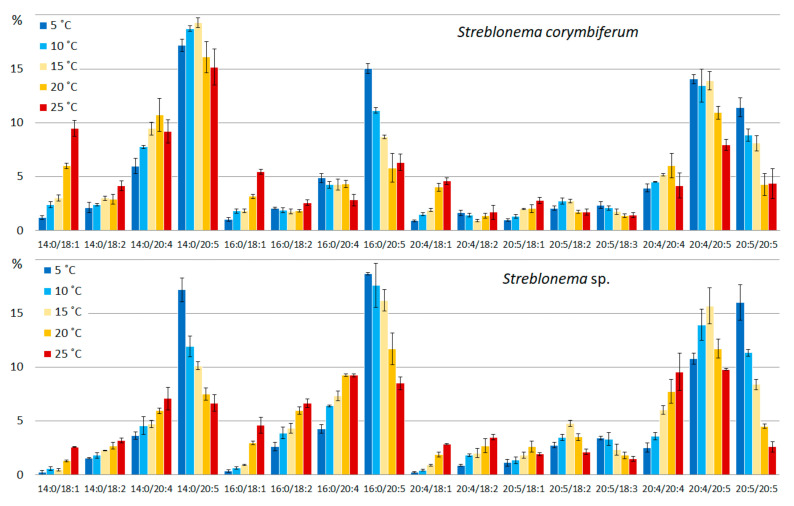
Phosphatidylcholine molecular species composition (% of total PC) of *Streblonema corymbiferum* and *Streblonema* sp. at various temperatures.

**Figure 7 marinedrugs-20-00428-f007:**
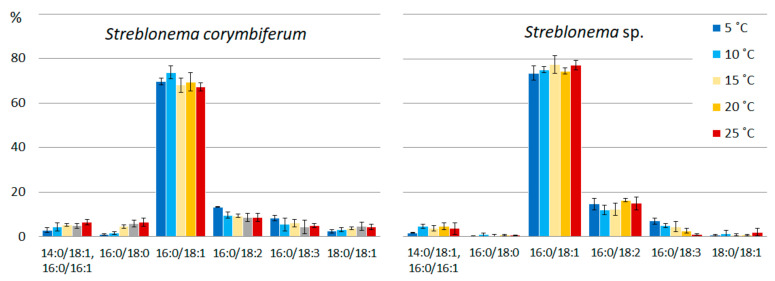
Phosphatidylinositol molecular species composition (% of total PI) of *Streblonema corymbiferum* and *Streblonema* sp. at various temperatures.

**Figure 8 marinedrugs-20-00428-f008:**
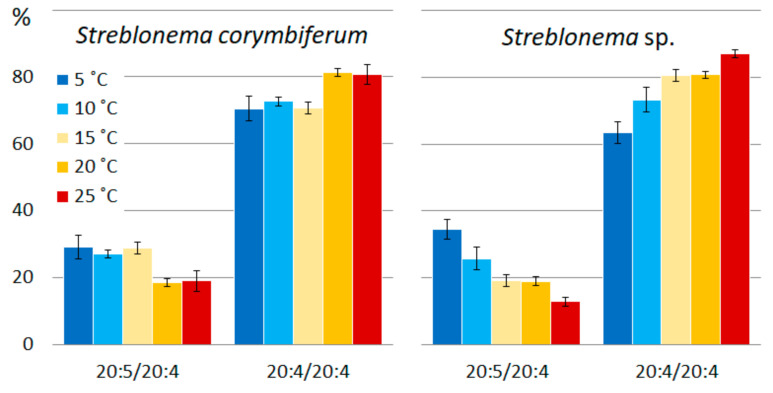
Phosphatidylhydroxyethylglycine molecular species composition (% of total PHEG) of *Streblonema corymbiferum* and *Streblonema* sp. at various temperatures.

**Figure 9 marinedrugs-20-00428-f009:**
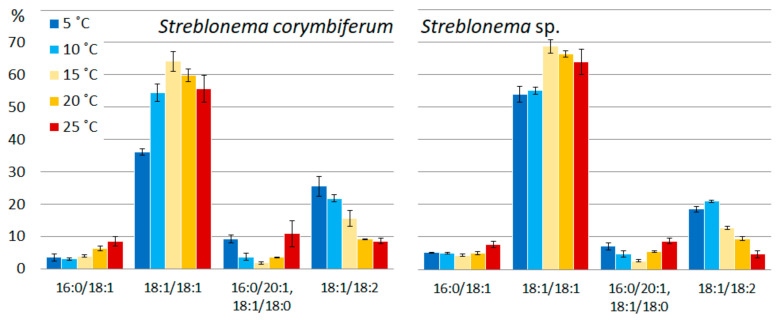
Diacylglyceryl-N,N,N-trimethyl-homoserine molecular species composition (% of total DGTS) of *Streblonema corymbiferum* and *Streblonema* sp. at various temperatures.

**Figure 10 marinedrugs-20-00428-f010:**
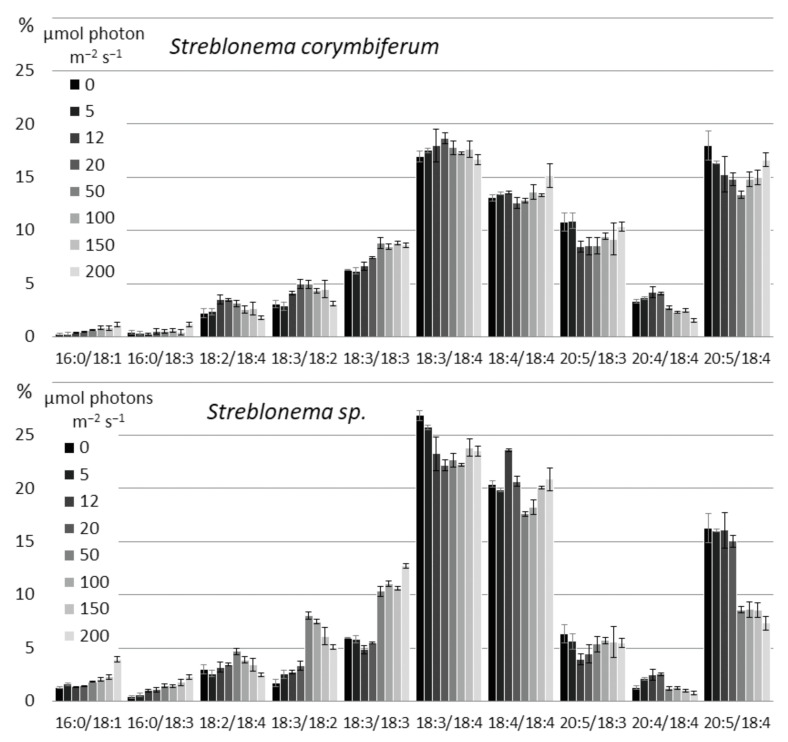
Monogalactosyldiacylglycerol molecular species composition (% of total MGDG) of *Streblonema corymbiferum* and *Streblonema* sp. at various light intensities.

**Figure 11 marinedrugs-20-00428-f011:**
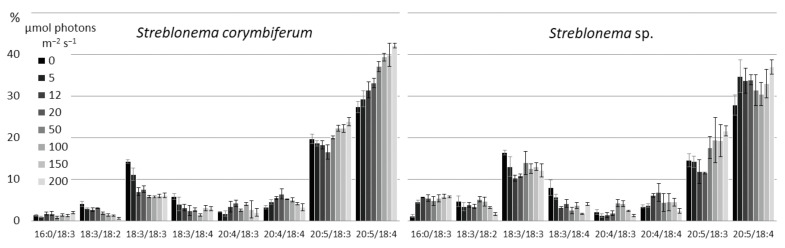
Digalactosyldiacylglycerol molecular species composition (% of total DGDG) of *Streblonema corymbiferum* and *Streblonema* sp. at various light intensities.

**Figure 12 marinedrugs-20-00428-f012:**
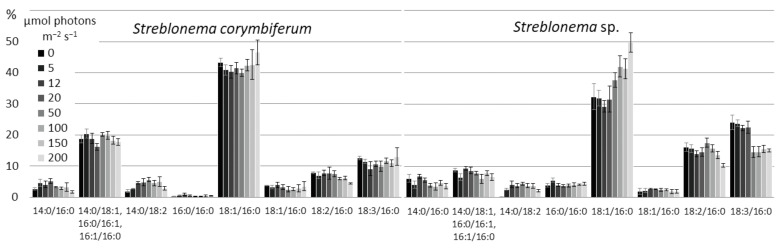
Sulfoquinovosyldiacylglycerol molecular species composition (% of total SQDG) of *Streblonema corymbiferum* and *Streblonema* sp. at various light intensities.

**Figure 13 marinedrugs-20-00428-f013:**
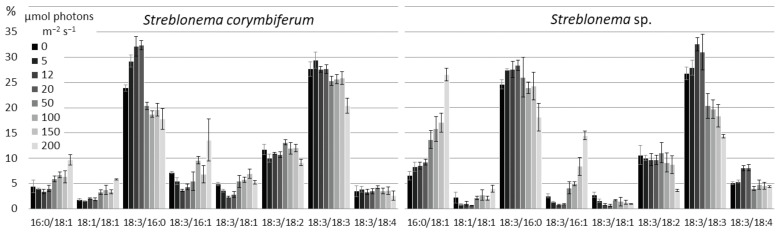
Phosphatidylglycerol molecular species composition (% of total PG) of *Streblonema corymbiferum* and *Streblonema* sp. at various light intensities.

**Figure 14 marinedrugs-20-00428-f014:**
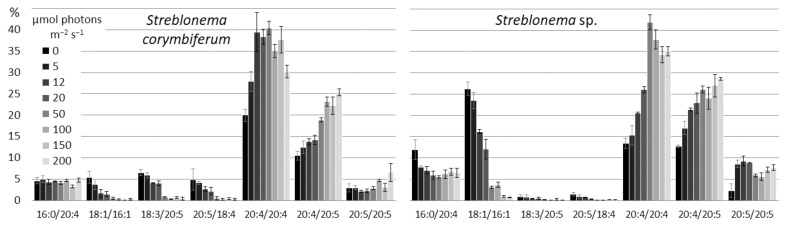
Phosphatidylethanolamine molecular species composition (% of total PE) of *Streblonema corymbiferum* and *Streblonema* sp. at various light intensities.

**Figure 15 marinedrugs-20-00428-f015:**
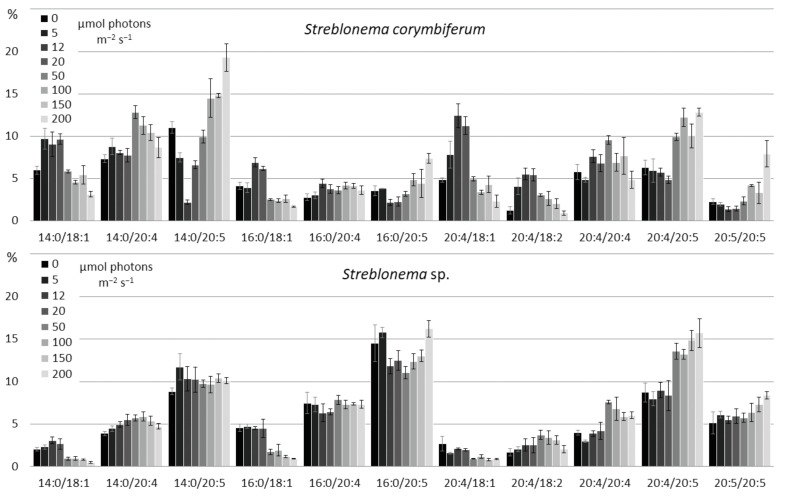
Phosphatidylcholine molecular species composition (% of total PC) of *Streblonema corymbiferum* and *Streblonema* sp. at various light intensities.

**Figure 16 marinedrugs-20-00428-f016:**
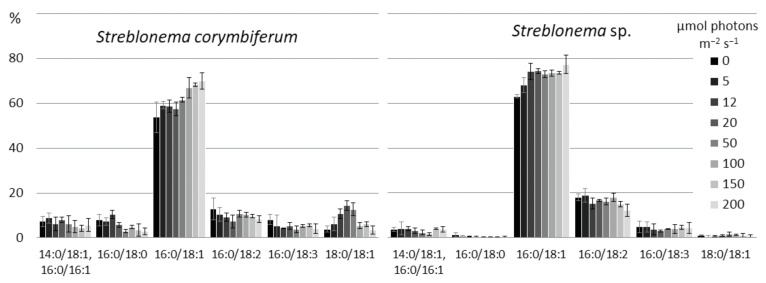
Phosphatidylinositol molecular species composition (% of total PI) of *Streblonema corymbiferum* and *Streblonema* sp. at various light intensities.

**Figure 17 marinedrugs-20-00428-f017:**
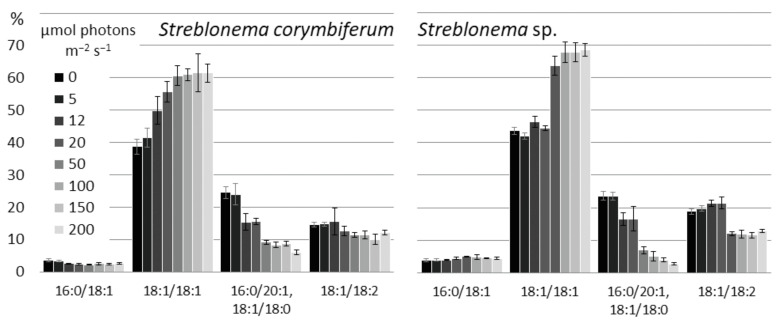
Diacylglyceryl-N,N,N-trimethyl-homoserine molecular species composition (% of total DGTS) of *Streblonema corymbiferum* and *Streblonema* sp. at various light intensities.

**Figure 18 marinedrugs-20-00428-f018:**
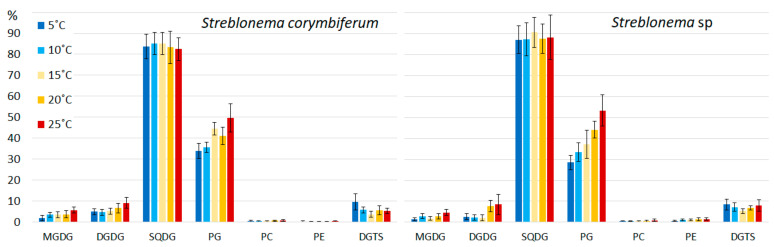
Level of chloroplast-derived molecular species of polar lipids (% of total lipid class) of *Streblonema corymbiferum* and *Streblonema* sp. at various cultivation temperatures.

**Table 1 marinedrugs-20-00428-t001:** Molecular species composition of polar lipids of *Streblonema corymbiferum and Streblonema* sp.

Lipid Class	Total Number of Molecular Species	Major Molecular Species
**Glycoglycerolipids**		
MGDG	113	18:2/18:4, 18:3/18:2, 18:3/18:3, 18:3/18:4, 18:4/18:4, 20:5/18:2, 20:5/18:3, 20:5/18:4
DGDG	63	16:0/18:3, 18:3/18:2, 18:3/18:3,18:3/18:4, 20:4/18:3, 20:4/18:4, 20:5/18:1, 20:5/18:2, 20:5/18:3, 20:5/18:4
SQDG	49	14:0/16:0, 14:0/18:1, 14:0/18:2, 14:0/18:3, 16:0/16:0,16:0/16:1, 16:1/16:0, 18:1/16:0, 18:2/16:0, 18:3/16:0
GlcADG	9	16:0/18:1, 16:0/18:2, 16:0/18:3, 18:1/18:1, 18:1/18:2, 18:1/18:3, 20:5/18:2, 20:5/18:3, 20:5/18:4
**Phosphoglycerolipids**		
PG	42	16:0/18:2, 16:0/18:1, 18:2/16:0, 18:3/16:0, 18:3/16:1, 18:1/18:1, 18:2/18:1, 18:3/18:1, 18:3/18:2, 18:3/18:3, 18:3/18:4
PE	42	14:0/20:4, 16:0/20:5, 16:0/20:4, 18:0/20:4, 20:0/20:4, 18:1/16:1, 18:3/20:5, 20:4/20:4, 20:4/20:5, 20:5/20:5
PC	85	14:0/18:1, 14:0/18:2, 14:0/20:4, 14:0/20:5, 16:0/18:1, 16:0/18:2, 16:0/20:4, 16:0/20:5, 20:4/18:1, 20:4/18:2, 20:4/20:4, 20:4/20:5, 20:5/20:5
PI	9	14:0/18:1, 16:0/16:1, 16:0/18:0, 16:0/18:1, 16:0/18:2, 16:0/18:3, 18:0/18:1
PHEG	3	20:4/20:4, 20:5/20:4
**Betaine lipid**		
DGTS	45	16:0/18:1, 18:1/18:2, 18:1/18:1, 16:0/20:1, 18:1/18:0, 18:1/19:1

MGDG, monogalactosyldiacylglycerol; DGDG, digalactosyldiacylglycerol; SQDG, sulfoquinovosyldiacylglycerol; GlcADG, glucuronosyldiacylglycerol; PG, phosphatidylglycerol; PE, phosphatidylethanolamine; PC, phosphatidylcholine; PI, phosphatidylinositol; PHEG, phosphatidylhydroxyethylglycine; DGTS, diacylglyceryl-*N*,*N*,*N*-trimethyl-homoserine. Numbers C:db indicates the number of carbon atoms (C) and double bonds (db) in the fatty acid chains. Molecular species whose content was higher than 5% from the sum of all molecular species of a given class (except GlcADG) in at least one of the samples are shown.

## Data Availability

Not applicable.

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
