# Peer review of "Effect of Temperature and Light Intensity on the Polar Lipidome of Endophytic Brown Algae Streblonema corymbiferum and Streblonema sp. In Vitro"

_marinedrugs, 2022, doi:10.3390/md20070428_

Round 1

Reviewer 1 Report

This work is a major effort at characterizing the lipidome of two macroalgae species with respect to changes in temperature and light intensity. The compositional change in temperature is expected as it has been observed in many different species in organisms from many different phyla. The general trend in greater unsaturation at lower temperatures is observed here. The lipid composition dependence on light is less obvious. It's not entirely clear what the authors thought they might observe and the interpretation of the data was not very compelling. 

The paper overall suffers from a lack of a hypothesis. It is very focused on characterization and does not address the importance of having this information. 

Some specific suggestions:

Lines 45-47: It's unclear to me why light changes are expected to be reflected in these particular membranes. Maybe a statement "If there are light-dependent changes in FA composition, they would be expected to appear in the thylakoid membranes as this is where photosynthetic light-dependent reactions occur" Based on my review of the light-dependent data the correlation between light intensity and FA composition is not very compelling.

Lines 48-49: What is light resistance?

Line 116: General comment on figures. labels are exceptionally difficult to read. Lot's of data presented and is difficult to follow. I spent significant amount of time trying to find 20:5/18:4 in figure 1 only to find out later that lipid is actually found in Figure 2. Placing the label at the end of the paragraph makes it difficult to tell which figure is being referred to. My other criticism is that there is so much data and the results narrative is exceptionally difficult to follow with so many acronyms and Carbon length:unsaturated bond notation. I wondered whether the general trends could be presented in a simpler format that wouldn't require all the data for all 10 lipid classes to be presented in their entirety. Each graph could certainly be available in the supplemental information.

The Discussion section reads very much like the results. Could the results and discussion be combined so that each lipid class is discussed separately. This might make the paper more clear. I also felt that the interpretation of data could be improved. I had a hard time understanding what the implications of the results were. Why is it important? What does it mean?

Lines 376-378: The authors seek to address how lipid composition can help maintain photosynthetic activity under conditions of low light and high light. The authors do not attempt to explain what causes the decrease in photosynthetic activity under each condition. It's not all related to FA composition. The causes that have the greatest influence on photosynthetic activity should be discussed and how FA composition could help the organism overcome these effects.

Line 444: Why use 200 µmoles as the "high" light? This is substantially lower than the 2,000 µmoles of full sunlight. Could differences be observed if the light intensity was truly "high"

Line 495: Why might temperature affect PI and PHEG least? What is their biological role?

The location of conclusions after Materials and methods seems like an odd fit. Is this the requirement of the journal?

Author Response

Response to Reviewer 1 Comments

Point 1: Lines 45-47: It's unclear to me why light changes are expected to be reflected in these particular membranes. Maybe a statement "If there are light-dependent changes in FA composition, they would be expected to appear in the thylakoid membranes as this is where photosynthetic light-dependent reactions occur" Based on my review of the light-dependent data the correlation between light intensity and FA composition is not very compelling.

Response 1: Since a change in the level of illumination primarily affects the efficiency of photosynthesis, it was therefore logical to assume that this would also affect the composition of lipids in photosynthetic membranes, especially since the function of these lipids in thylakoids is not limited to structural (DOI: 10.1007/978-90-481-2863-1_12).

Point 2: Lines 48-49: What is light resistance?

The term "light resistance" is indeed incorrect in this sentence; we have replaced it with "sensitivity".

Point 3:  Line 116: General comment on figures. labels are exceptionally difficult to read. Lot's of data presented and is difficult to follow. I spent significant amount of time trying to find 20:5/18:4 in figure 1 only to find out later that lipid is actually found in Figure 2. Placing the label at the end of the paragraph makes it difficult to tell which figure is being referred to. My other criticism is that there is so much data and the results narrative is exceptionally difficult to follow with so many acronyms and Carbon length:unsaturated bond notation. I wondered whether the general trends could be presented in a simpler format that wouldn't require all the data for all 10 lipid classes to be presented in their entirety. Each graph could certainly be available in the supplemental information.

Response 3: We've increased the font size on the Figures as much as possible for better visual perception. The 20:5/18:4 molecular species is present in both Figure 1 and Figure 2. References to figures have been moved to the beginning of the paragraph describing this figure. Presenting Figures in a simpler format will not be correct (for example, labels in the form of gross formulas will not give any information about a particular FA and its sn-position in the lipid molecule. The presence of molecular species with the same gross formulas (for example, 18:2/18:4 and 18:3 / 18:3) complicates the situation even more. We did not include graphics in Supplemental information, since the text will be difficult to read, and the Figures make it possible to visually assess the changes. It is not possible to reduce the number of Figures, since all the submitted Figures are discussed.

Point 4: The Discussion section reads very much like the results. Could the results and discussion be combined so that each lipid class is discussed separately. This might make the paper more clear. I also felt that the interpretation of data could be improved. I had a hard time understanding what the implications of the results were. Why is it important? What does it mean?

Response 4: We have not combined Results and Discussions as it is preferable for the journal to provide them separately. The significance and importance of the results obtained are indicated in the Conclusion.

Point 5: Lines 376-378: The authors seek to address how lipid composition can help maintain photosynthetic activity under conditions of low light and high light. The authors do not attempt to explain what causes the decrease in photosynthetic activity under each condition. It's not all related to FA composition. The causes that have the greatest influence on photosynthetic activity should be discussed and how FA composition could help the organism overcome these effects.

Response 5: We have added a brief introductory description of the effect of light on a photosynthetic organism. We discuss how changes in FA composition can help to overcome various effects of light, briefly explaining what processes they may be involved in.

Point 6: Line 444: Why use 200 µmoles as the "high" light? This is substantially lower than the 2,000 µmoles of full sunlight. Could differences be observed if the light intensity was truly "high"

Response 6: Our preliminary experiments showed, that at light intensity higher 200 μmol m-2 s-1 Streblonema sp. showed chlorosis and growth reduction after one month. We do not exclude that higher linght intensisty may effect the lipidome composition of the studied algae, however, we are confident that full sunlight is deadly for these algae. In nature both species growth endophytically in macroalgae, where they are exposed to very low light. Experiments perfomed on other endophytic ectocarpalean alga (Ectocarpus siliculosus) showed that light higher 100 μmol m-2 s-1 inhibited the growth (DOI: 10.1016/j.aquaculture.2022.738526).

Point 7: Line 495: Why might temperature affect PI and PHEG least? What is their biological role?

Response 7: The structural role (and, accordingly, the regulation of the physicochemical properties of membranes by changing the FA composition) of these lipids is not the main one. PI is a precursor of phosphoinositides, involved in signaling processes, vesicular transport and reorganization of the cytoskeleton (DOI: 10.1152/physrev.00028.2012). Little is known about the function of PHEG. This lipid has been detected in gametes of algae, and since it contains only С20 PUFA, it has been suggested that this lipid may be an acyl donor for pheromone production and participate in the fertilization of brown algae (DOI: 10.1016/S0176-1617(11)81999-1).

Point 8 The location of conclusions after Materials and methods seems like an odd fit. Is this the requirement of the journal?

Response 8: Yes, it is the requirement of the journal.

Reviewer 2 Report

Review Comments:

Title: Effect of Temperature and Light Intensity on the Polar Lipidome of Endophytic Brown Algae Streblonema corymbiferum and Streblonema sp. in Vitro

Remarks:

The manuscript entitled on “Effect of Temperature and Light Intensity on the Polar Lipidome of Endophytic Brown Algae Streblonema corymbiferum and Streblonema sp. in Vitro” as described by Song et al is written well. The research is ready to be publish in its current form but can be considered after the minor amendments. The content of the research is enough and supported by enough reference evidence. I would recommend this paper to be accepted after minor revision. Following are my specific comments to further polish the manuscript:

General Comments -

Ø  Title – Acceptable.

Ø  Abstract – Abstract need to be improved. Obtained results need to be summarized. The first few lines introductory lines can be added. A last few line the future prospective need to be included.

Ø  Keywords – Acceptable.

Ø  Introduction – Intro part is enough, but some part needs to be improved. Please refer similar articles to revise it properly. Novelty of this work? Explain? Following references could be helpful and can be referred to improve the introduction.

https://doi.org/10.1007/s13399-020-00786-y

https://doi.org/10.2166/wst.2021.195

Ø  Materials – Need to be reproducible. Can be acceptable in present form. As per my opinion the construction of manuscript cab be done as follows:

-          Abstract

-          Introduction

-          Materials and Methods

-          Results

-          Discussions

-          Conclusion

Following references will be helpful in explanation of temperature treatment, light treatments, lipid extraction and GCMS analysis of microalgae.

https://doi.org/10.1016/j.biteb.2021.100696

https://doi.org/10.15414/jmbfs.2020.9.4.671-674

https://doi.org/10.1007/s13399-019-00548-5

Ø  Results – Result and discussion part is acceptable. How the algae species were isolated and identified? Authors can provide kinds of identification images for the Check the references cited. Please check the superscripts carefully.

Ø  Conclusion – Need to be improved. Conclusions need to be summarized with obtained results. Last lines add some future prospective of the respective study.

Minor Specific Comments –

  1. Check the numbering of subtitles.
  2. Follow the author guidelines properly.
  3. Include high quality graphs and figures.
  4. More recent references need to be included from 2020 to 2022.
  5. Some grammatical mistakes should be corrected throughout the manuscript.

In the end I would like to summarize that above corrections should be done to improve the overall quality of the manuscript. 

Author Response

Response to Reviewer 2 Comments

Point 1: Abstract – Abstract need to be improved. Obtained results need to be summarized. The first few lines introductory lines can be added. A last few line the future prospective need to be included.

Response 1: Abstract corrected.

Point 2: Introduction – Intro part is enough, but some part needs to be improved. Please refer similar articles to revise it properly. Novelty of this work? Explain? Following references could be helpful and can be referred to improve the introduction.

https://doi.org/10.1007/s13399-020-00786-y

https://doi.org/10.2166/wst.2021.195

Response 2: Novelty of this work explain (Lines 40-52, 59-64)

Point 3:  Materials – Need to be reproducible. Can be acceptable in present form. As per my opinion the construction of manuscript cab be done as follows:

Abstract

Introduction

Materials and Methods

Results

Discussions

Conclusion

Following references will be helpful in explanation of temperature treatment, light treatments, lipid extraction and GCMS analysis of microalgae.

https://doi.org/10.1016/j.biteb.2021.100696

https://doi.org/10.15414/jmbfs.2020.9.4.671-674

https://doi.org/10.1007/s13399-019-00548-5

 Response 3: The construction of the article is done according to the requirements of the journal.

Point 4: Results – Result and discussion part is acceptable. How the algae species were isolated and identified? Authors can provide kinds of identification images for the Check the references cited. Please check the superscripts carefully.

Response 4: The methodology of isolation of the endophyte is described in Material and Methods section (see subsection 4.1). Because the paper is focused on the analysis of lipidome, but not taxonomy of the endophytes, the description of the identification of the endopytes is seems exceed in the main text. We have provided this information in the Supplimentary materials.

The superscripts checked and corrected.

Point 5: Conclusion – Need to be improved. Conclusions need to be summarized with obtained results. Last lines add some future prospective of the respective study.

 Response 5: Conclusion corrected.

Minor Specific Comments –

    1. Check the numbering of subtitles - Сhecked.
    2. Follow the author guidelines properly –
    3. Include high quality graphs and figures -
    4. More recent references need to be included from 2020 to 2022 – We did not find suitable references from 2020 to 2022.
    5. Some grammatical mistakes should be corrected throughout the manuscript –Performed